# HomeMonitor: An Enhanced Device Event Detection Method for Smart Home Environment

**DOI:** 10.3390/s22239389

**Published:** 2022-12-01

**Authors:** Meng Zhao, Jie Chen, Zhikai Yang, Yaping Liu, Shuo Zhang

**Affiliations:** Cyberspace Institute of Advanced Technology, Guangzhou University, Guangzhou 510006, China

**Keywords:** smart home devices, privacy protection, device event detection, device event signature

## Abstract

As more and more smart devices are deployed in homes, the communication between these smart home devices and elastic computing services may face some risks of privacy disclosure. Different device events (such as the camera on, video on, etc.) will generate different data traffic during communication. However, the current smart home system lacks monitoring of these device events, which may cause the disclosure of private data collected by these devices. In this paper, we present our device event monitor system, HomeMonitor. HomeMonitor runs in the OpenWRT system and supports complete event monitoring for smart home devices. HomeMoitor solves the problem that machine learning models for detecting device events do not scale flexibly. It uses the network packet size and the direction of the device event for unique identification during training. When detecting, it only needs to get the packet size and timestamp and then query the policy table for signature matching to control the device events. We evaluated the effectiveness of HomeMonitor, and the experiments show that the match rate of our method is 98.8%, the false positive rate is 1.8%, and the detection time is only 16.67% for PINBALL. The results mean that our method achieves the balance of applicable protocol scope, detection performance, and detection accuracy.

## 1. Introduction

With the continuous improvement of smart home systems, more and more users choose to use smart home devices in their homes, and this growth is continuing. More comprehensive privacy protection and better service are two critical requirements for today′s smart home systems.

However, as far as we observe the phenomenon, in order to provide the appropriate services to users, more device privacy data needs to be acquired. Research on privacy and security for smart homes is underway. Smart home systems are generally divided into three main bodies in architecture: smart home devices, traffic transmission links, and smart home cloud platforms. According to the different research subjects, the current research work also can be divided into three areas. 

This work targets the traffic transport link privacy leakage problem. Most of the mainstream smart home systems adopt the three-level control architecture of “App-Cloud–Device.” Devices do not provide a device status display function when collecting or transmitting data. Users′ home gateways also lack monitoring devices. Therefore, the user is not clear about the function and real-time operation status of the smart home device, but the device can transmit the collected data without the user′s awareness or authorization [1,2,3]. The uploaded data may well be maliciously stolen, leading to serious consequences of family privacy leakage. Monitoring the event of smart home devices, increases the transparency of the system and allows users to know the activity status of their devices, having good application value for protecting the privacy of smart home users.

Unfortunately, we find that smart home device event monitors face challenges to fine-grained detection, real-time, and flexibility. Much of the existing research [4,5,6,7,8,9,10,11,12,13,14,15,16,17,18,19,20,21,22,23,24,25,26,27,28,29,30,31,32,33,34,35] addresses device-level identification and partial address of device-event-level identification. Some research [4,5,6,7,8,9,10,11,12,13,14,15,16,17,18,19,20,21,22,23,24,25,26,27] uses the statistical characteristics of device traffic combined with machine learning. Other research [28,29,30,31,32,33,34,35] make use of some simple but effective features called device event signatures, which usually consist of the exact value of the packet length and the packet direction, etc. 

Fine-grained detection: This is because device event detection implies that the identification goal is not only to classify the encrypted traffic into devices but also to classify the encrypted traffic into device events. Here device event specifically refers to various functional activities of the device, such as the camera′s taking pictures, recording, viewing videos, etc. Device event identification is more fine-grained and requires more demanding identification methods. 

Flexibility: Machine learning-based methods cannot flexibly add new event types to the model because the machine learning model has been trained before. It also cannot flexibly adapt to changes in device behavior at the network layer, such as firmware updates of the device, changes in the processing flow of device functions, and differences in the environment in which the device is located, all of which can lead to changes in the packet characteristics of the same device behavior at the network layer, requiring retraining of the entire model in order to identify new device behaviors. 

Real-time: In machine learning-based methods, the accuracy and real-time of device behavior recognition depend on the acquisition of classification features and the design of classification models. In particular, the acquisition and calculation of features at the data stream level need to wait for the end of the entire stream before they can be performed, so high real-time performance cannot be achieved. In the device event signature-based method, it cannot achieve a balance between the real-time and the scale of the protocol. For example, PINGPONG [32] first proposed device event signatures at the network layer, but its signature extraction method for TCP data flow does not apply to UDP data flow. The subsequently proposed PINBALL [33] is more compatible with the protocol but requires additional information statistics and computational consumption.

Based on the above discussion, we choose the current state-of-the-art solution, PINBALL [33], to improve. In order to expand the scope of the application of device event signatures and reduce computing consumption, DESEND (Device Event Signature Extraction and Detection Method) was presented [36]. It is a novel device signature extraction and detection method. The main ideas are as follows. Firstly, we extract all high-frequency packets in the event trigger windows as the key part of the signature. Secondly, we utilize the maximum time interval of signature packets with a flexible number of matches to eliminate the effect of re-transmitted packets. 

In this work, we designed a smart home monitoring system, HomeMonitor, and deployed DESEND+ on it, which is based on our original method, DESEND. Our contribution is as follows:We designed a flow monitoring system called HomeMonitor to achieve the entire process of device event monitoring. For smart home devices based on OpenWRT and DESEND+ to verify the effectiveness of our proposed method. Our system HomeMonitor is designed and implemented based on real IoT situations, which are valuable in practical applications.We presented DESEND+, an enhanced event signature extraction, and detection method. The detection speed is faster than PINBALL [33]. At the same time, it has a wider range of applications than PINGPONG [32] which means it can be applied to both TCP and UDP protocols. Furthermore, we added a device event control function to intercept device events by intercepting critical packets. Machine learning methods based on data flow features for this function are difficult to implement, while statistical methods based on packet features do not focus on this.

This paper is organized in the following structure. Section 2 introduces the background and motivation. Section 3 introduces the Design of HomeMonitor. Section 4 introduces the evaluation and implementation of HomeMonitor. Section 5 introduces the related work. Section 6 introduces the conclusion and future work. 

## 2. Background and Motivation

The existing smart home systems lack monitoring of smart home device events. Some unexpected device events may leak users′ privacy information. This section introduces threats faced by smart homes, the current issue of device event detection in smart homes, and our idea.

### 2.1. Threat to Smart Home

The development of the smart home system can be divided into three stages: single-device intelligence, multiple-devices intelligence, and whole-house intelligence stage. The development of device intelligence and the growing demand for cloud services have led to more frequent communication between devices and the cloud. Now, we are in the third stage. Mass information interaction increases the risk of leakage of users’ privacy, such as images, voice, and so on.

Zhou et al. [1] tested the interaction between app, cloud, and device in a smart home system. They found that malicious attackers could obtain a device ID through the firmware of the user′s device. The device ID can be used to bind the device and attacker′s account to achieve device hijacking. The red path in Figure 1 shows the direction of data flow for this attack. Data from the device can be sent to the attacker′s app.

Some security incidents have also caused users′ concerned about the privacy disclosure of smart homes. Researchers [2] activated Amazon Alexa through a voice activation program to continuously record sound without the user′s knowledge. In another case, users can watch video recordings from other users′ devices in the app under certain special usage scenarios. We show those privacy leak attacks in Figure 1. In the blue path, the service cloud continues to collect data from smart home devices without the user′s knowledge. Existing methods are difficult to use DESEND+ against such attacks unless the smart home system is targeted for improvement.

We believe that cloud platforms and malicious attackers can continuously collect data from smart home devices through the normal process of the system. Since existing smart home systems lack monitoring of device events, users could not aware of the possible potential threats. It is important to reduce the possibility of privacy leaks by creating a mechanism for IoT device event detection in the smart home.

### 2.2. Problem Analysis

There are two common methods of device event detection, machine learning-based detection and device event signature. The disadvantage of the first method is that it is impossible to collect features in flexible and real-time, and the model needs to be continuously updated to cope with changes in the characteristics of equipment events. Therefore, the current state-of-the-art method is device event signature [32,33].

In Figure 2, the signature of method PINGPONG [32] includes (l,d), and the packets in the signature are ordered. l denotes the size of the packet, and d denotes the direction of the packet. In Figure 3, we show some operations in the PINGPONG signature extraction process. In the training phase, PINGPONG removes re-transmission packets from the data flow. As shown in Figure 3, the number of packets with sizes 1277 and 1276 is almost the same as the number of events triggered after processing. These packets will be added to the device event signature. If the device event generates a UDP data stream, we cannot remove the re-transmission packets. Packets of size 1276 and size 1277 will not be added to the signature. It means the ON and OFF event of the Sengled-light-bulb will be indistinguishable.

Similarly, we show the event signature of PINBALL in Figure 2. We assume that the number of events triggered is N and the packets in the signature appear in at least 0.9N windows. PINBALL [33] calculates the proportion of occurrences of each packet as P. Then it calculates the similarity between the packet distribution Q(x) in the window and the device event signature P(x).

Figure 4 shows the processing flow of PINBALL [33] in the detection phase. It first constructs a detection window with a length of 10 s. After the calculation is completed, the detection window moves to the right for 1 s and continues to repeat the calculation operation. Therefore, the calculation consumption in the detection phase is related to the number of packets and the duration of the data flow. In the real smart home environment, the device’s idle time is much longer than the device’s active time. During detection, PINBALL will generate more computing consumption in the device’s idle time.

### 2.3. Our Idea

Our main idea is as follows:We use device event signature as the basic method of device event detection.Only the size of the packets is collected as a signature element feature, and there is no order between the signature elements, which makes the device event signature robust.We use a transport protocol-independent mechanism to handle re-transmitted packets.In the detection phase, we use packet matching to complete event detection. The detection calculation consumption is independent of the duration of the data flow, so it has a low calculation consumption.We control the critical packets to achieve the control function for device events.

According to PINBALL, some packets will appear in most device events that trigger windows. We divide these packets into two sets: the key-packet set (key) and the high-frequency-packet set (high). The packets in the key set will appear in each trigger window, and the packets in the high setting will appear in at least most trigger windows.

In the feature selection, we only consider the packet size feature. When we build the event signature, we have no special requirements for the number of packets, so we can include more packets in the signature. We give an example of the ON-event of the TP-LINK bulb device in Table 1. Compared with PINGPONG [32], there are more packets in the device event signature extracted by DESEND+. In general, we think that the feature of packet size has enough specificity.

During detection, re-transmitted packets will increase the number of duplicate event reports. We give an example in Figure 5. After the TP-LINK bulb color event is triggered, the packets in the set (58, 287, 317) will appear stably. The red packets are re-transmission packets in the network flow. The detection program detects whether the network flow contains data packets with the size in the set (58, 287, 317). Three event reports will be generated, and the red report is a duplicate report. In repeated reports, the maximum time interval of packets is usually large, so in the training phase, we extract the maximum time interval of packets in the signature. If the next target packet is not detected within the maximum time interval, we will clear the matched part so that duplicate reports will not be generated. Just such as PINGPONG, the calculation consumption of DESEND+ in the detection phase is only related to the number of packets, which avoids excessive calculation consumption.

PINGPONG uses the sequence of packets and the overall duration of signatures to eliminate the impact of re-transmission packets. However, the signature extraction policy of PINGPONG restricts its application on UDP data flows. PINBALL uses the distribution of packet size in the detection window as the event signature, and fewer re-transmission packets have little impact on the distribution. However, the disadvantage of PINBALL is that the calculation consumption in the detection phase is high.

We compare the existing works in four dimensions: calculate consumption, protocol, scalability, and robustness. Scalability reflects the ability to easily adjust the model to detect new events after a change in device event. HomeSnitch [27] uses a single machine learning model for device event detection, which is less scalability. The event signature-based methods have better scalability. The results of the comparison are shown in Table 2.

## 3. HomeMonitor

### 3.1. Overview of HomeMonitor

In this section, we present our device event monitor system, HomeMonitor. HomeMonitor runs in the OpenWRT system and supports complete event monitoring for smart home devices. Figure 6 shows the overview of HomeMonitor. 

HomeMoitor addresses the problem that machine learning models for detecting device events do not scale flexibly. It uses the network packet size and direction of device events to uniquely identify, which is the network event signature of a device event. The signatures of different events combine to form the event signature file of the device, and when the network traffic of an event changes, the network packet signature of the event is retrieved and added. The detection and matching module only needs to get the packet size and timestamp and then query the policy table for signature matching. According to the matching result, the current packet should perform the operation, which includes permit, deny and ask. These operations are defined in advance by the user through monitor rules. The output is processed separately by action control. Permit means that the packet passes, deny means that the packet is dropped, and a warning is sent to the user, and ask means that a query is sent to the user.

#### 3.1.1. Event Signature Extraction and Training Module

The current effective training data collection method is proposed by Trimananda et al. [1]. First, we connect the device to the gateway and open the traffic collection program at the gateway. Then we repeat the trigger device event and record the timestamp of the trigger. Finally, we use the DESEND+ method to extract the device event signature.

In HomeMonitor, we use the device event description file to save the device event signature. A single device event description file consists of all device event signatures for that device.

#### 3.1.2. Event Signature Detection and Matching Module

The event detection program receives every packet from the smart home device from the Open vSwitch. Then determine whether the source IP address or destination IP address of the packet is related to the smart home device and the related packets enter the subsequent matching process.

For each event signature, the detection program maintains two queues corresponding to the set of key-packet and the set of high-frequency-packet in the signature. Each packet size within the signature corresponds to a flag, which defaults to 0. If a packet size within a signature is matched, the corresponding flag is set to 1. After packet matching is finished, it will get the matching result corresponding to different operations, and the matching result will be sent to the control module for processing.

#### 3.1.3. Event Control Module

After receiving the action message from the event signature detection and matching module, the event control module can issue the corresponding ACL (Access Control List) control rule to the Open vSwitch, which will release or restrict the data flow according to this rule.

Users can customize the monitor rules, which will be converted into entries in the policy table. In addition, the user can also modify the policy table by modifying the monitor rules. When the action message received by the module from the upstream module is “notify the user to make a decision,” the module will feed this information back to the user and wait for the user to make a decision.

### 3.2. Event Signature Extraction and Training Module

#### 3.2.1. Packet Capture Framework

As shown in Figure 7, The packet capture framework in HomeMonitor consists of three parts: wlan2, Open vSwitch, and Ryu controller. wlan2 provides wireless network access for devices. Open vSwitch uploads the device′s traffic to the Ryu controller based on the user-configured flow table. The device event detection program in the Ryu controller detects events based on the device event signature. After the detection is completed, the packet is sent to the bridge br0.

#### 3.2.2. Signature Extraction and Training

In the training phase, repeatedly trigger the device events N times by manual or automatic means, saving the traffic file and triggering the timestamp. The triggered event was expressed as $E_{i},\;i\in \left\{1,2,3,\ldots ,N\right\}$, and the triggered timestamp was set as $T_{i},\;i\in \left\{1,2,3,\ldots ,N\right\}$. For each event $E_{i}$, DESEND+ extracts the traffic features within the $\left[T_{i,},T_{i}+t\right]$ time window, t is set to 10s in the experiment. DESEND+ extracts the packet size Pl as part of the device event signature, the timestamp Pt for the subsequent checks. 

There are two main sets of device event signatures extracted by DESEND+: the key-packet set (key) and the high-frequency-packet set (high). After an event was triggered, in t time, packet size Pl will be extracted. If Pl appears N times in all windows, we add Pl to the key-packet set. If Pl appears at least 0.9N times in windows, we add Pl to the high-frequency packet set. If two packets are subtracted from each other with a difference of 1, such as their packet sizes $Pl_{1}-Pl_{2\;}\;$= 1. We set $Pl_{1},\;Pl_{2}$ as $\left(Pl_{1},Pl_{2}\right)$. If $Pl_{1},Pl_{2}$ appears at least 0.9N times, we add $\left(Pl_{1},Pl_{2}\right)$ to the set of high-frequency-packet set.

Packets may often be re-transmitted in the complexity of the network situation. It causes inconsistencies between the number of event signature matches and actual event triggers. DESEND+ eliminates this effect in event signatures by limiting the match interval between two adjacent packets. Maximum match interval mechanism can work well on both TCP and UDP packets.

As shown in Table 3, we list the signatures extracted by DESEND+. DESEND+ extracts packets from the key-packet set and the high-frequency-packet set then calculate the time interval of the adjacent matching packets in two sets and obtain the maximum interval time (set as *max_time*) and minimum occurrences of high-frequency packets (set as *min*). The two sets of extracted data by DESEND+ are not in order, so the change of packet order in the real data will not affect the validity of the signature.

In the detection phase, there may be a variety of reasons that will cause certain packets in the set the key-packet set (key) and high-frequency-packet set (high) not to appear, such as the training set containing background traffic. To ensure a certain degree of fault tolerance, DESEND+ adds the parameters *fix* and *high fix*. The length of the key packet set is *key length*, and the key-packet set (key) is considered to be complete when the number of key-packet matches reaches *key length–fix.* The number of matches required is *min–high fix* for the high-frequency-packet set (high). The algorithm of event signature extraction is shown in Algorithm 1.

**Algorithm 1.** Event signature extraction **INPUT**: input pkgs = *{*
$P_{1}$, $P_{2}$,…, $P_{n}$
*}*    Timestamp file: T = *{*
$T_{1}$, $T_{2}$,…, $T_{n}$
*}*    event_trigger_time # Number of events triggered when collecting data**OUTPUT**: Event signature1. **get**
*counter_list*
**#** counter_list save the pkg len in [Ti,Ti + 10s]2. **# example:**
*counter_list* = [[T1][P1.len,PK.len],[T2][][],...]3. **get** *pkg_occurance_num* # pkg_occurance_num saves the number of trigger intervals with B4. # *pkg_occurance_num* = [[len:P1.len][num:10],...]5. **for each** *pkg_num* **in**
*pkg_occurance_num*6.  **if** *pkg.num == event_trigger_time*7.    *event_signature.key.append(pkg.len)*8.  **if** *pkg.num* >= *event_trigger_time-5*9.    *event_signature.high.append(pkg.len)*10. **endfor**11. **for each** *interval* **in** *interval list*12.  *temp_min* = Number of occurrences of elements in set event_signature in interval13.  **if**
*temp_min* < *min*14.     *event_signature.min = temp*15. **endfor**16. **for each** *interval*
**in**
*interval list*17.  *temp_max_time* = The maximum time between two packets in a signature in interval18.  **if** *temp_max_time* > *max_time*19.     *event_signature.max_time = max_time*20. **endfor**

As shown in Table 4, after signature extraction and training, a signature file would be generated.

### 3.3. Event Signature Detection and Matching Module

#### 3.3.1. Policy Table

Policy configuration provides users with configurable monitoring options based on device event signatures. Users configure monitoring rules based on monitoring requirements. Specific information is shown in Table 5. 

#### 3.3.2. Detection

The main idea in the detection phase is to detect if there are packets in the data flows that can match the policy table. The event signature detection program keeps two separate matching sequences for the set. A match is considered successful if the required number of matches is reached without exceeding *max_time*. Otherwise, the match is considered to have failed. The detection sequences will be cleared. In addition, the matching packets should send messages to the event control module according to the action in the policy table.

A device may have multiple events, each packet with the same IP as the device will be matched against all event signatures, and it will be detected at the same time. There are two cases in which the matching sequences for each event signature will be cleared: one is no new packets are added to the detection sequences within the *max_time, and* the other is the event about this device matches successfully. The algorithm of event signature detection is shown in Algorithm 2.

**Algorithm 2.** Detection and matching **INPUT**: input pkgs = *{*
$P_{1}$, $P_{2}$,…, $P_{n}$
*}*    policy table: *F* = *{*
$F_{1}$, $F_{2}$,…, $F_{n}$
*}***OUTPUT**: Event_Match1. **for each**
$P_{i}$
**in**
pkgs
**do**2.   **for each**
Fi **in** *F*3.     **if**
Pi**.***ip*
**==**
$F_{i}$**.***ip*
**do**4.        **if**
*L(*Pi*)*
**in**
$F_{i}$.*key*
**and**
*L(*Pi*)*
**not in**
Fi*.keylist*
**do**5.           Fi*.keylist.append(L(*Pi*))*6.           Fi*.lastpkt_time = T(*Pi*)*7.        **endif**8.        **if**
*L(*Pi*)*
**in**
$F_{i}$*.high*
**and**
*L(*Pi*)*
**not in**
Fi*.highlist*
**do**9.           Fi*.highlist.append(L(*Pi*))*10.          *lastpkt* = *T(*Pi*)*11.       **endif**12.    **endif**13.  **endfor**14.  **for each**
Fi
**in**
*F*15.    **if**
*L(*$F_{i}$*.keylist) >= L(*$F_{i}$*.key)–fix*
**and**
*L(*$F_{i}$*.highlist) >= L(*$F_{i}$*.key)–high fix*
**do**16.       **Event_Match**
*[*Fi*.name].append(*Fi*.lastpkt_time)*17.       **clearall**
*F.keylist, F.highlist, F.lastpkt_time*18.       **break**19.    **endif**20.  **endfor**21.  **for each**
Fi
**in**
*F*22.    **if**
*T(*Pi*)–*Fi*.lastpkt_time <*
Fi*.max_time*23.       **clear**24.       **clear**
Fi*.keylist,*
Fi*.highlist,*
Fi*.lastpkt_time*25.    **endif**26.  **endfor**

### 3.4. Event Control Module

#### 3.4.1. Monitoring Rule

Users configure monitoring rules according to monitoring requirements. The event signature is generated by the security service vendor or device manufacturer. Policy configuration generates the final monitoring policies based on monitoring rules and device event signature. Event detection applies different monitoring policies based on device classes. The event detection results are displayed to the user through the event detection report.

As shown in Table 6, a monitoring rule consists of two parts: the matching field and action field. The match field includes device id, device event, and valid time. 

#### 3.4.2. Action Control

We designed device event control based on event detection. The goal of device event control is to control device events at the network layer while not affecting other events of the device. The action field contains two optional monitoring actions: log and intercept, which means to record information or drop the packet. Since the set of keys in the event signature is steadily occurring, we refer to the packets in this set that can affect the device event in progress as critical packets. We continuously intercept the critical packets while the device event is in progress, which will prevent the completion of the event, such as the transmission of video, etc. The duration of the interception will be set by the user. We will verify the feasibility of device event control in the experimental section. 

## 4. Evaluation and Implementation

### 4.1. The Implementation of HomeMonitor

#### 4.1.1. Experimental Environment

We tested DESEND+ in a real environment with the topology shown in Figure 8 and the information of the gateway shown in Table 7. We place four devices in the lab. The evaluation of the HomeMonitor system was divided into three parts.

#### 4.1.2. Device Event Signature Extraction

We connected three camera devices in the lab to the wireless hot spot of the gateway. We disabled the video upload feature and the human detection feature for each camera, both of which generate network traffic unrelated to viewing video and are triggered in an uncontrollable manner. We connect our smartphone to the campus network WiFi and trigger the camera to view the video in the corresponding smart home app, repeating 20 times for 20 s each time and 2 min between each trigger.

Table 8 shows the device event signatures for the three cameras in the lab. In particular, Xiaomi smart camera 2K uses UDP packets to transmit video traffic.

#### 4.1.3. Device Event Detection

We watch the videos of the Xiaomi smart camera 2K, EZVIZ-C2C camera, and Lenovo R1 camera on the APP. The mac address of the Xiaomi smart camera 2K is 60:7E:A4:30:70:60, and the detection record is shown in Table 8.

When performing device event detection, the EZVIZ-C2C camera and the Lenovo R1 camera were able to report very quickly after watching a video. Xiaomi smart camera 2K needs to report a few seconds after watching the video. This is related to the overall duration of the device event signature.

### 4.2. The Evaluation of Detection

In this section, we analyze the calculated consumption of DESEND+ in the detection phase. To verify the validity of various parameters in the event signature, we conduct experiments on them separately. We evaluated the detection accuracy and performance of the DESEND+ method using publicly available datasets and compared it with existing event signature-based methods.

#### 4.2.1. Evaluation Metrics

From the perspective of accuracy and time consumption, we evaluate the effectiveness of DESEND+. Accuracy is made up of two elements: the match rate and the number of false positives. The match rate indicates the ability of the method to correctly detect device events. A false positive is a duplicate or incorrect recording of a device event. Time consumption is determined by calculation consumption. It also can be divided into two components: packet feature extraction consumption and matching calculation consumption.

Assuming the file contains N packets for a total of S seconds, the calculation consumption for event detection in this file is D. In the packet feature extraction step, the calculation consumption to perform one comparison is assumed to be a. m is the number of features collected from each packet. n is the number of comparisons. It is related to the size and number of device event signatures. The calculation consumption at this step is a∗N∗m∗n.

The calculation consumption during the match calculate step depends on the design of the method and is related to the key-step calculation, assuming that one key-step calculation is d, the number of key-step is T, T is related to the number of packets N the duration of the data flow S, and the event signature of the device, which can be calculated by the function Cal_key. The time consumption of the final calculation is d∗Cal_key(N,S).

Combining the two steps, the calculated consumption of event detection is shown in Equation (1). The parameters a, m, n, d are constants, Cal_key is linearly related to N, so the overall computational complexity is O(n).
(1)$$\begin{array}{c} D=a*N*m*n+d*Cal\_key\left(N,S\right) \end{array}$$

#### 4.2.2. Simulation Settings

To evaluate DESEND+, we perform event signature extraction and detection experiments on two public datasets: PINGPONG and MonIoTr [6]. In the PINGPONG dataset, we perform accuracy evaluation and time consumption evaluation. PINBALL can also extract event signatures for TCP and UDP flows, so we use PINBALL as the baseline for comparison. In the MonIoTr dataset, we perform signature extraction experiments to verify the effectiveness of signature extraction on UDP data flows.

For accuracy evaluation, we evaluate the effect of *max_time* and *fix* parameters on the accuracy. For time consumption evaluation, we compare the time consumption of DESEND+ and PINBALL in the detection phase.

Dataset 1: PINGPONG dataset. PINGPONG was the first work to propose a network layer signature of a device event. They collected a dataset of smart home device events. Devices in this dataset include cameras, switches, bulbs, etc. It includes a PCAP file and a timestamp.txt file. PCAP file contains 100 times active trigger for each device event, timestamp.txt file contains the start timestamp of each device trigger, and the device trigger interval is set to 131 s. We distinguish device events that are triggered at different times based on timestamps. We use the data of device events in the following path as the training set: /evaluation-datasets/local-phone/standalone. At the same time, the data in the following path is used as the test set: /evaluation-datasets/local-phone/smarthome. The above data contains 23 types of device events. Device Signatures extracted by DESEND+ are also released for public access [36]. Here is the link to the dataset: PingPong: Dataset | UCI Networking Group.

Dataset 2: MonIoTr dataset. The MonIoTr dataset contains traffic generated by 55 different smart home devices, devices in this dataset include speakers, doorbells, tv, etc. It contains both TCP-based and UDP-based device events. DESEND+ verifies the validity by extracting the signature of the UDP-based device event from the MonIoTr dataset. In the dataset, the traffic generated by each device event is saved as a separate PCAP file, and the traffic packets generated by different device events are classified in different folders, and we use the folder names as labels for the traffic samples. Here is the link to the dataset: IoT Information Exposure (IMC ′19)–Mon(IoT)r Research Group (neu.edu)

#### 4.2.3. Accuracy

We perform signature extraction experiments on devices using UDP protocol in the MonIoTr dataset, and some of the results are shown in Table 9. There is no limit on the time to trigger in the MonIoTr dataset.

We use the PINGPONG datasets to verify the accuracy of DESEND+. With respect to the matching success rate, DESEND+ is able to match 98.8% of the device events, indicating that DESEND+ can effectively identify the device events in the data flows with a low error rate. With respect to the false positive rate, DESEND+ generates 1.8% duplication due to packet re-transmissions.

The impact of the two-evaluation metrics on device event detection is different. The match rate is a direct reflection of the methods the under-reporting rate, where an under-report means that a device event is not being detected correctly. False positive means the detection program generates some duplicate reports.

We conducted event detection experiments on 23 device events. Among the 23 devices, the detection trigger times of 19 device events are inconsistent with the actual trigger times. We show the detection accuracy of some device events in Figure 9. The worst one is the ON-OFF event of the sengled-bulb, where nine event triggers were not detected.

In Figure 10, the first four devices match better in DESEND+ with the larger difference in ring-alarm, while the last three devices match worse in DESEND+ with the sengled-bulb performing worse in detecting the ON-OFF event. Overall, the difference between DESEND+ and PINGBALL in terms of matching success ratio is not significant.

In the DESEND+, parameters max_time and fix are very important. We performed experiments with them separately.

The max_time parameter is the maximum time interval between two adjacent matching packets in the device event signature. It is used for eliminating duplicate matches generated by re-transmitted packets. In practice, we increase the maximum time interval by 0.5 s as max_time, which increases the redundancy of event detection.

As shown in Figure 11, a total of 139 false positives were reported by the five devices when max_time was not enabled. With max_time enabled, only 11 false positives were reported by the five devices. In particular, the TP-LINK bulb does not report any false positives in the test.

Among related work, PINBONG filters re-transmitted packets with sequence numbers of TCP packet headers, making it unsuitable for UDP data flows. PINBALL uses the percentage of occurrence of high-frequency packets in a statistical interval to eliminate duplicate matches, yet it requires more additional computation. DESEND+ reduces computing consumption while supporting both TCP and UDP protocols.

Other parameters in the event signature extracted by DESEND+ are fix,high fix, which affects the matched number required for the event signature.

The adaptability of the matched number ensures the robustness of the device event signature. The fix,high fix in the device event signature is set to 0 by default. As shown in Table 10, the ON and OFF event of the seven devices need to adjust the values of fix and high fix in the detection phase.

Figure 12 shows the variation of the matching rate of the seven devices with the fix parameter enabled. Among the seven devices, the fix parameter has a greater impact on the TP-LINK bulb device. Taking the ON-event of the TP-LINK bulb as an example, the set of key-packet of the device event signature extracted from the datasets is (46, 58, 71, 198, 227, 309, 520, 1049). However, in the test file, the TP-LINK bulb with address 192.168.1.246 does not send or receive packets of size 309, so the key-packet set cannot be matched successfully without adjusting the fix value.

#### 4.2.4. Time Consumption

The above gives an analysis of the accuracy of DESEND+. In real applications, it is also necessary to consider time consumption, which is positively related to computational consumption. We divide it into two steps, the feature extraction step, and the match calculation step. 

In the feature extraction step, according to Equation (1), m represents the number of features. We use the size and time of the packets as features. Thus, the number of features m is 2. The number of comparisons n is related to the number of key-packet and high-frequency-packet in the device event signature.

In the match calculate step, DESEND+ has two key steps d. One is to judge if the size of the packet matches the one in the key-packet set or the high-frequency-packet set. The other one is to calculate if the interval time of two matching packets is less than the max_time. d is related to the number of packets N to be detected in the data flow, independently of the data flow duration S.

The ON-event of the wemo-plug device is taken as an example. The signature of the ON-event is shown in Table 11.

The test file has 15,484 packets related to wemo-plug, the number of collected features is 2, the number of packets in the event signature is 4, the number of calculations in the two key steps is 15,484, and the calculation consumption D is as follows:(2)D=a∗15484∗2∗4+d∗15484∗2

We conducted event signature extraction and event detection experiments on 23 devices with PINBALL and DESEND+. The time.perf_counter function is used to obtain the current timestamp of the system at the beginning and end of the detection function. The difference between the two is used to calculate the time consumption of the calculation phase. The average detection time for the five rounds is shown in Figure 13. 76.418 s per round for PINBALL and 9.926s per round for DESEND+.

DESEND+ vs. PINBALL is shown in Table 12. The DESEND+ method shows more false positives than PINBALL, but the detection time of DESEND+ has a greater advantage over PINBALL. The detection efficiency is improved by about 80.6%.

### 4.3. The Experiment of Device Event Control

#### 4.3.1. Device Event Control Experiment

In the event control experiment of the devices in Table 13, we take the Lenovo R1 camera as an example. Respectively intercepting data packets of sizes 125, 184, 234, and 1488, the program reports a large number of re-transmission packets. As shown in Figure 14, the smart Lenovo App is unable to view the video, and the connection progress will be stuck at 89%. As mentioned above, this data stream is critical for the event to view video. 

The same interception experiments were performed separately for the other device event. The results as shown in Table 13. For the fluorite C2C camera, an intercepted packet of size 52, which is sent to the device, will make the camera cannot view the video. The table is marked as 0. However, a packet of size 52, which is the device sent out, has no effect on viewing the video. The table is marked as 1. For the Xiaomi smart camera 2K, which uses UDP, intercept data packets of size 38, 40, 56, 60, and 64 have no impact on viewing the video, while packets of size 96 and 1060 have an effect on viewing the video.

#### 4.3.2. Event Detection Report

The results of the device event detection are presented to the user on a web page. The information reported is shown in Figure 15. log_time is the time when the device event occurred, and device_event represents the detected device event. Device_id is the mac address of the device. The detection results are listed in chronological order.

## 5. Related Work

With the rapid development of the Internet of Things (IoT), many works investigate how to mitigate the security privacy threats of IoT. Through the communication and activity status of IoT devices, some threat information can be obtained, and corresponding responses can be made. Related work can be divided into two categories: device event detection and device traffic monitoring.

### 5.1. Device Event Detection

Since data from existing smart home devices are often encrypted [6,7], detecting device events means detecting them from encrypted traffic. The research problem focuses on two aspects: (1) the selection of effective traffic characteristics and (2) the development of a detection method that takes into account the handling of retransmitted packets and unknown class packets.

As shown in Table 14, there are two types of methods to detect devices and device events from network traffic: machine learning-based detection methods and statistical classification-based detection methods.

Machine learning-based detection methods: Many works [4,5,6,7,8,9,10,11,12,13,14,15,16,17,18,19,20,21,22,23,24] use the traffic features of devices and train supervised machine learning models for user event detection. These methods have high detection accuracy. Some works use unsupervised models [25,26] to detect unknown classes of devices and events in network traffic.

Regarding device event detection, Abbas et al. [24] proposed a multi-stage user behavior detection scheme that applies device detection, device state detection, and device state classification. They combined multiple device states to extract user behavior. Oconna et al. [27] proposed a system called HomeSnitch. It collects flow features of device events (e.g., maximum packet size and average packet size in the flow) and uses these features to train a random forest model to detect device events.

Statistical feature-based detection methods: Statistical features of traffic are used in these works [28,29,30,31,32,33,34,35] for monitoring, such as the set of DNS request domain names [28], the set of TCP port numbers [29], and the TF-IDF values of TLS packets [30]. 

Device event signature [32,33] is an important work in statistical feature-based detection methods with good accuracy and flexibility. PINGPONG [32] system, which was presented by Trimananda et al., could extract packet-level signatures from device event-triggered traffic. It detects device events by determining whether device event signature packets are present in network flows. However, the limitation of PINGPONG is that it does not apply to the UDP protocol. Duan et al. [33] present improved signature extraction and detection method, PINBALL. The device event signature extracted by PINBALL contains all high-frequency packets in *t* seconds after the device event is triggered. PINBALL uses the distribution of the proportion of packets as the device event signature.

For both types of device event detection methods, there is little difference in accuracy. In the training phase, machine learning-based methods usually contain a single detection model and are less scalable than device signature-based methods. In the detection phase, machine learning-based methods need to wait for device events to complete before extracting traffic features, so some methods cannot perform real-time detection. Event signature-based methods do not need to wait for device events to complete, so it is easier to achieve real-time detection.

### 5.2. Device Traffic Monitoring

Most work in this category depends on packet header information, especially for MUD-based systems [37]. With MUD files, users can restrict the communication range of IoT devices, and the MUD manager matches traffic packets based on IP, port, and protocol information. Wang et al. [38] refined the description granularity of the MUD file based on the IFTTT automation procedure and added some descriptive information such as time interval, packet number, packet sequence, etc. However, it has some limitations since MUD files cannot describe device event information.

Other works have focused on how to mitigate the risk of privacy disclosure by reducing the upload of privacy data to cloud services. Chi et al. [39] present the PFIREWALL system, where they analyzed and obtained the minimum data required by the home automation program. Without modifying the smart home architecture, the device can only upload the minimum data required by the home automation program to the cloud platform. Xu et al. [40] present a privacy protection framework for cloud platforms. They propose an F&F filtering component that filters redundant behavior records uploaded to IFTTT and blurs the features of the uploaded data. These works can protect user privacy at the data level but are only applicable to some smart home platforms, such as SmartThings and IFTTT.

In general, existing research on traffic monitoring focuses on device communication range restriction, malicious traffic detection, and data leakage reduction [41].

## 6. Conclusions and Future Work

In this paper, we present a novel device event signature extraction and detection method, DESEND+. Then, in order to detect events in a real smart home environment, we built a flow monitoring system, HomeMonitor. According to the experimental results, the match rate and the false positive rate of DESEND+ are both higher than PINBALL [33], while the detection time of DESEND+ is only 16.67% of PINBALL [33]. In addition, the DESEND+-based HomeMonitor supports both TCP and UDP data flows. In future work, the stability of event signatures can be further verified, such as whether the signatures change over time. The implementation of Homemonitor can also be optimized to reduce the network latency caused by event detection, then further improving the event control mechanism. 

## Figures and Tables

**Figure 1 sensors-22-09389-f001:**
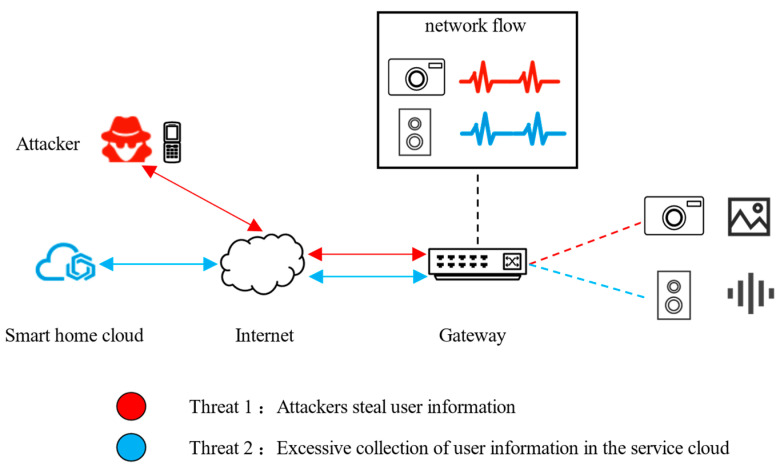
Smart home privacy threats.

**Figure 2 sensors-22-09389-f002:**
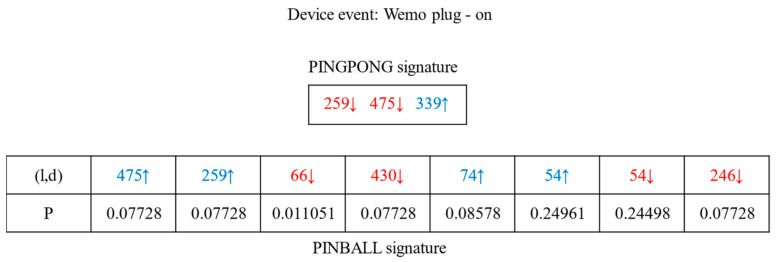
Wemo-plug on-signature.

**Figure 3 sensors-22-09389-f003:**
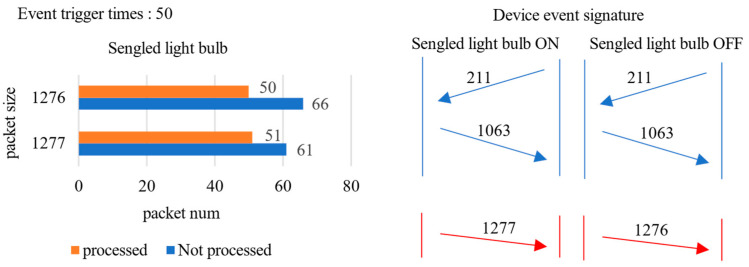
Signature detection process in PINGPONG.

**Figure 4 sensors-22-09389-f004:**
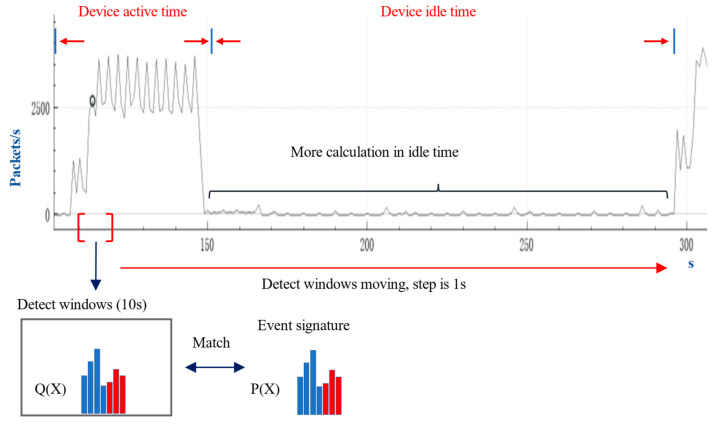
Signature detection process in PINBALL.

**Figure 5 sensors-22-09389-f005:**
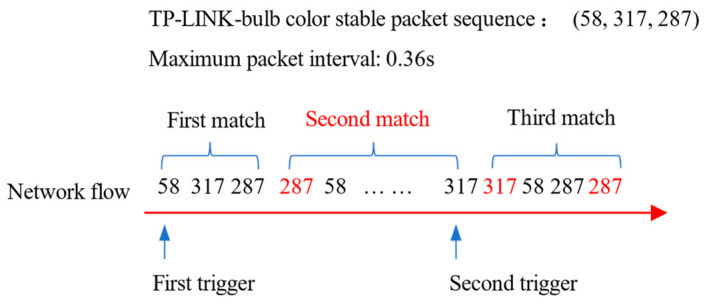
Signature detection process in DESEND+.

**Figure 6 sensors-22-09389-f006:**
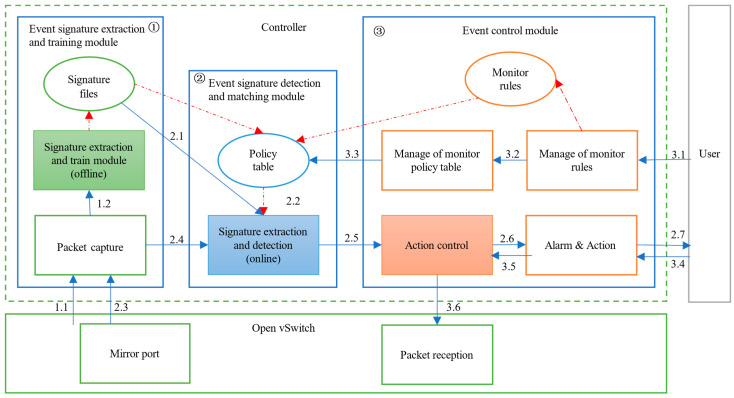
HomeMonitor architecture.

**Figure 7 sensors-22-09389-f007:**
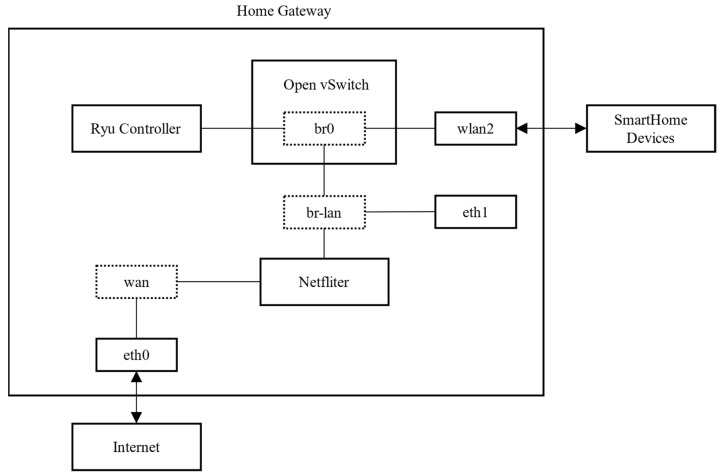
Packet capture framework.

**Figure 8 sensors-22-09389-f008:**
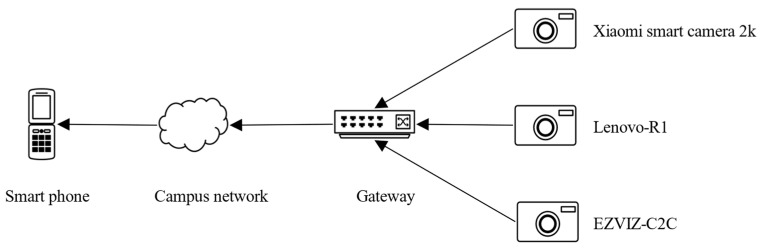
DESEND+ in a real environment with the topology.

**Figure 9 sensors-22-09389-f009:**
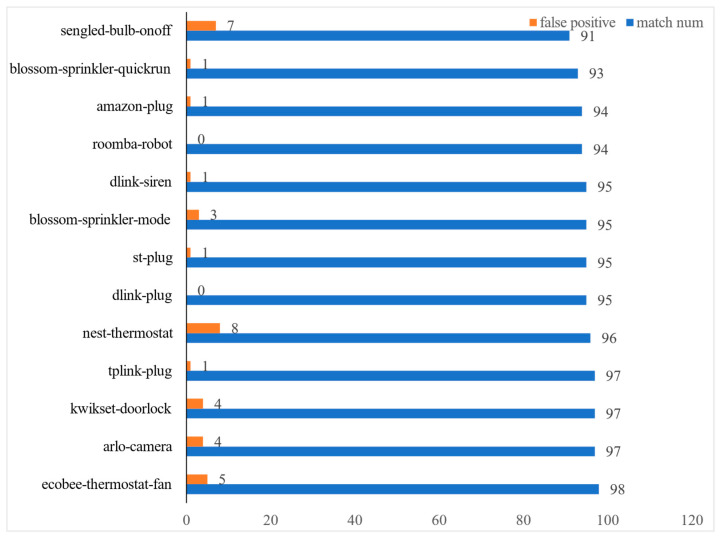
Device matching experiment.

**Figure 10 sensors-22-09389-f010:**
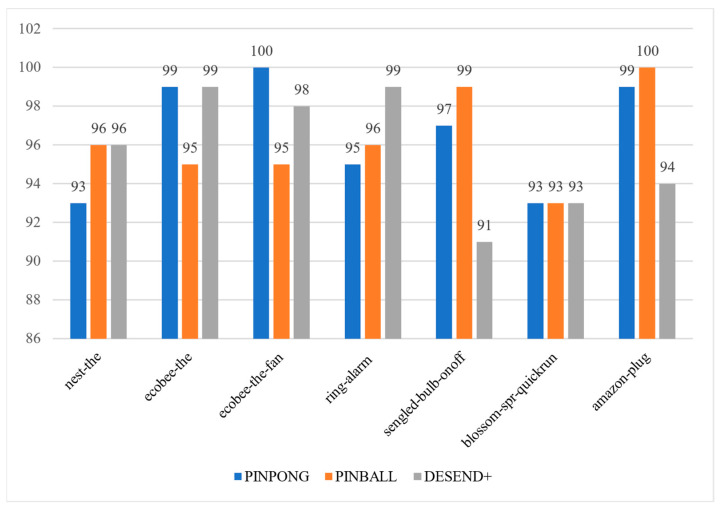
Event detection experiment.

**Figure 11 sensors-22-09389-f011:**
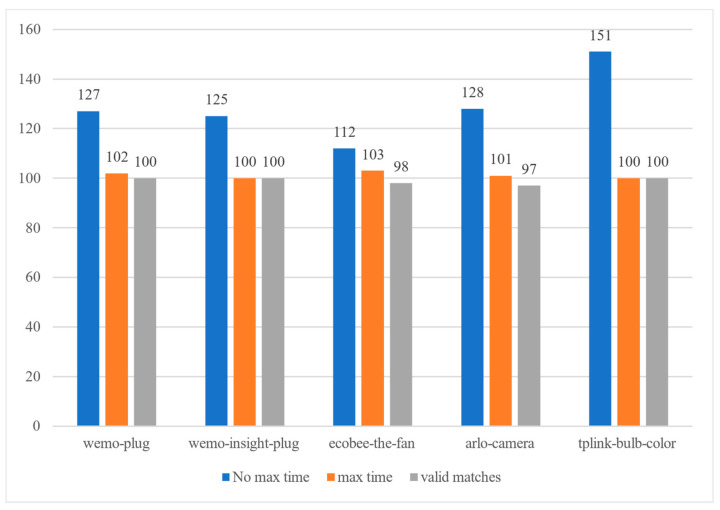
Max_time parameter experiment.

**Figure 12 sensors-22-09389-f012:**
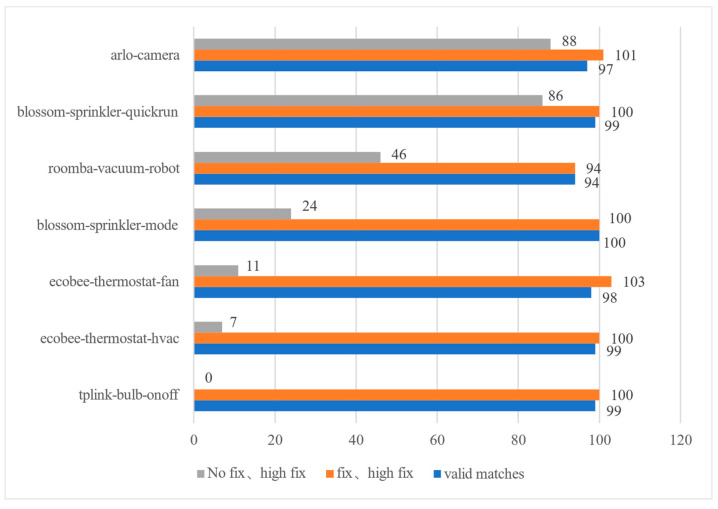
Fix parameter experiment.

**Figure 13 sensors-22-09389-f013:**
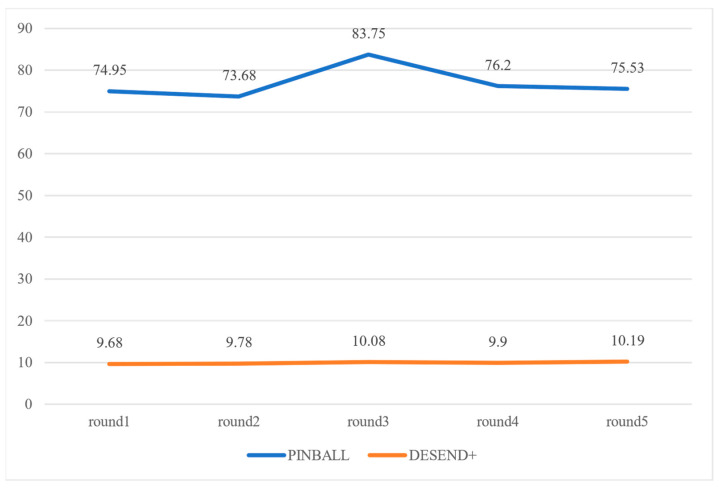
Time consumption comparison.

**Figure 14 sensors-22-09389-f014:**
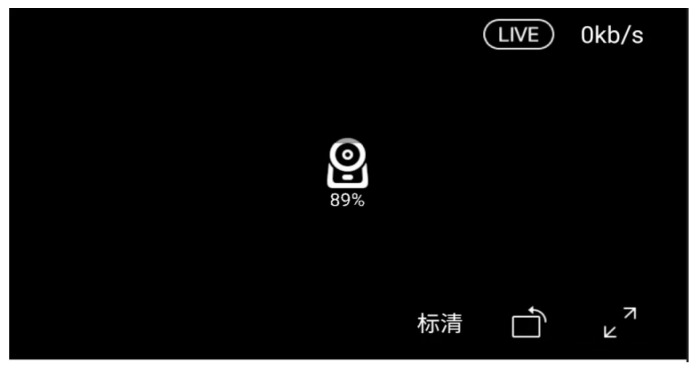
Lenovo R1 View the video unsuccessful.

**Figure 15 sensors-22-09389-f015:**
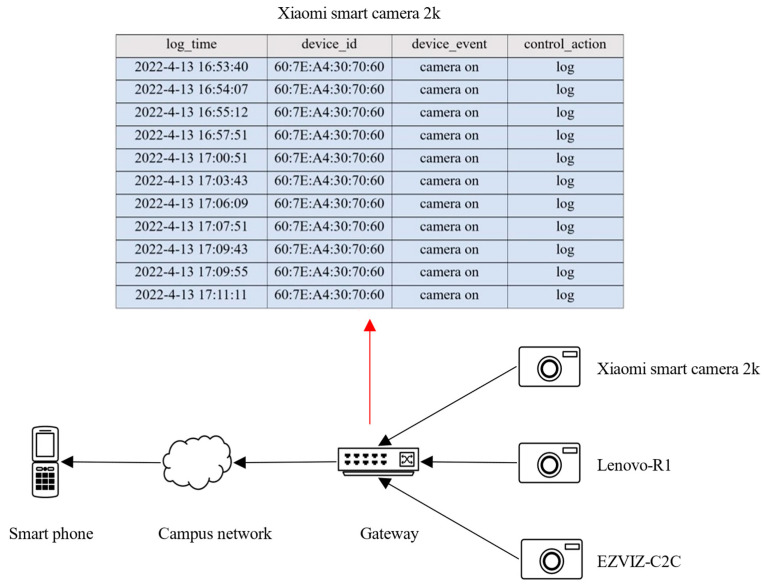
Device event detection.

**Table 1 sensors-22-09389-t001:** TP-LINK bulb ON-event.

	Packet Size Contained in the Signature
DESEND+	key: (46,58,71,198,227,309,520,1049) high: (627,1311,1454)
PINGPONG	198, 227

**Table 2 sensors-22-09389-t002:** Alternative approaches.

Work	Feature	Protocol	Fast Detection	Events Control
HomeSnitch	complex	TCP	Yes	No
PINGPONG	simple,	TCP	Yes	No
PINBALL	simple	TCP, UDP	No	No
DESEND+	simple	TCP, UDP	Yes	Yes

**Table 3 sensors-22-09389-t003:** Description in device event signature extract reproduced with permission from [J. Chen, Y. Liu, S. Zhang, Z. Guo, B. Chen and Z. Han], [DSC]; published by [IEEE], [2022].

Component	Description
name	device event name
key	key-packet set
high	high-frequency-packet set
*min*	minimum number of occurrences in the set of high-frequency-packet set
*max_time*	maximum interval between two adjacent matching packets
*fix*	key-packet-set matching number adjustment value
*high fix*	High-frequency-packet set matching number adjustment value

**Table 4 sensors-22-09389-t004:** Device event signature file.

Keyword	Description
manufacturer	device manufacturer
name	device name
version	File Version
event num	Number of device events
event list	Device event signature list

**Table 5 sensors-22-09389-t005:** Policy table.

Device (Mac/IP)	Event	Signature			Action
		Key	High	…	
(allure-speaker) 192.168.1.7	android_wan audio_ON	83, 74, 338, 60, 267	343, 54	…	deny
	android_wan audio_OFF	83, 74, 338, 267, 60	-	…	ask
(roku-tv) 192.168.1.9	android_lan remote	60, 97, 66, 74, 1514	80, 75, 378, 373, 192, 117, 519, 1396, 607, 208, 244, 214	…	permit
	android_wan remote	74, 66	97, 1514, (85, 86)	…	deny
(lightify-hub) 192.168.1.15	android_lan_ ON	260, 196, 235, 108	93	…	permit
	android_lan_ OFF	260, 196, 235	93, 108	…	permit

**Table 6 sensors-22-09389-t006:** Monitoring rule.

	Keyword	Description
match field	rule_id	Rule id
	Mac/IP	Device Mac/IP address
	device_event	
	start_time	Effective start time
	end_time	Effective end time
action field	control action	Log/Drop

**Table 7 sensors-22-09389-t007:** Gateway information.

CPU	i5-8400
ram	8 GB
OpenWRT	19.07
dnsmasq	2.8.5
hostapd	v2.10-devel

**Table 8 sensors-22-09389-t008:** Device event signature in the lab.

Device	Event	Signature	Protocol
Lenovo-R1 camera	watch video	52, 60, 104, 106, 125, 184, 234, 1488	TCP
Xiaomi smart camera 2K	watch video	38, 40, 56, 60, 64, 96, 1060	UDP
EZVIZ-C2C camera	watch video	52, 60, 72, 120, 1440	TCP

**Table 9 sensors-22-09389-t009:** MonIotr datasets experimente.

Device	Event	Key	High	Min	Max Time
allure-speaker	android_wan audio_ON	83, 74, 338, 60, 267	343, 54	1	41.26
	android_wan audio_OFF	83, 74, 338, 267, 60	-	2	43.24
roku-tv	android_lan remote	60, 97, 66, 74, 1514	80, 75, 378, 373, 192, 117, 519, 1396, 607, 208, 244, 214	12	18.19
	android_wan remote	74, 66	97, 1514, (85, 86)	3	30.53
lightify-hub	android_lan_ ON	260, 196, 235, 108	93	1	13.58
	android_lan_ OFF	260, 196, 235	93, 108	1	13.38

**Table 10 sensors-22-09389-t010:** Fix parameter adjustment.

Device Event	Fix (ON)	High Fix (ON)	Fix (OFF)	High Fix (OFF)
TP-LINK bulb	1	0	1	0
ecobee-thermostat-havc	1	0	1	0
ecobee-thermostat-fan	1	0	1	0
blossom-sprinkler-quickrun	0	3	0	0
blossom-sprinkler-mode	0	1	1	0
alro-camera	0	1	0	1
roomba-vacuum-robot	1	0	1	0

**Table 11 sensors-22-09389-t011:** Signature extracted by DESEND+.

	Value
key	(246, 259, 430, 475)
high	()
min	0
max_time	0.56366491317749023
fix	0
high fix	0

**Table 12 sensors-22-09389-t012:** Comparison between DESEND+ and PINBALL.

	Match Rate	False Positive Rate	Detection Time (s)	UDP Support
PINBALL	98.4%	0.08%	76.418	yes
DESEND+	98.8%	1.8%	9.926	yes

**Table 13 sensors-22-09389-t013:** Device event control experiment result.

Device Event	Intercepted Packets	Result
Lenovo R1-View the video	125, 184, 234, 1488	unable to view video
Fluorite C2C-View the Video	1440, 120, 72, 52(0), 60	unable to view video
Fluorite C2C-View the Video	52(1)	able to view video
Xiaomi smart camera 2K-View the Video	96, 1060	unable to view video
Xiaomi smart camera 2K-View the Video	38, 40, 56, 60, 64	able to view video

**Table 14 sensors-22-09389-t014:** Comparison of two types of methods.

Method		Accuracy	Demand	Feature
Machine learning classification	Supervised Learning	High	Precise traffic classification	High accuracy
	Unsupervised Learning	Middle	Unknown traffic classification Traffic law extraction Feature validity evaluation	Identify unknown traffic
	Deep Learning	High	Precise traffic classification Automatic feature extraction	High accuracy, Unsupervised
Statistical classification	High	Precise traffic classification	Well-designed methods could realize real-time classification	Statistical classification

## Data Availability

The data presented in this paper will be made available on request via the author’s email with appropriate justification.

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
