# Peer review of "HomeMonitor: An Enhanced Device Event Detection Method for Smart Home Environment"

_sensors, 2022, doi:10.3390/s22239389_

Round 1
Reviewer 1 Report
In this manuscript, the authors propose a HomeMonitor system to monitor and control the behavioral flow of smart home devices. In HomeMonitor, they leveraged DESCEND, an enhanced device event signature extraction, and detection method. It is faster and more applicable than existing works. In addition, the author built a traffic monitoring system running in the OpenWRT system. This paper could have a meaningful impact on the Internet of Things (IoT) subfield.
Strength:
1. The HomeMonitor system proposes in this paper is feasible and has practical application value.
2. The author provides a comprehensive description of the system, including a detailed framework figure, algorithmic process, etc.
3. Compared with the related existing works, this paper has the fastest detection time. Moreover, it supports both TCP and UDP data flows for device event detection.
4. The paper presents complete and detailed experiments and evaluations, and the experimental environment is based on real IoT devices.
Weakness:
1. The reference format is messy. For example, there are italic, bold, non-bold, etc.
2. The paper's figures are unclear, such as figures 11, 13, 15, etc. It is recommended to replace the figures with higher resolution.
3. The HomeMonitor system is limited in innovation, mainly using existing technology, which is DESCEND with no clear improvement.
4. There is an issue with the organization of the paper. The technical method of DESCEND is the existing work. Therefore, the contents of DESCEND need to be introduced in the section preliminaries.
5. Reduce contents on the DESCEND method and focus on the author's work.
Author Response
We feel great thanks for your professional review work on our article. As you are concerned, there are several problems that need to be addressed. According to your nice suggestions, we have made some corrections to our previous draft, point-by-point responses to the reviewer are listed below this letter. All changes to the manuscript has been highlighted.

Reviewer 2 Report
1. The authors should explain the importance of the system implemented ( MonitorHome detection) . I think they are interessted only to detect event on a smart environment however it's vey important to detect eventual attacks or intrusion in smart home.
2. In the end of the paper, authors present two big shutters machine learning and statistical classification-based detection are not enough explained. You can apply machine learning that have a high performance to detect intrusion.. So what's the performance of the proposed method face to machine learning?
3. The authors present an enhanced event signature extraction and detection method. What's the utility of detecting event if they don't look for possible intrusion and attacks ?
4. the detection speed is faster than PINBAL How you can confirm that is more faster ?
5. Quality of figures 13, 15 must be improved also the tables didn't cut them (table 9, 5,etc)
6.I suggest to replace Figure 14 by an architecture of the whole system
7. It's recommended to give a detailed information of the two datasets used in this work, the description is not enough.
Author Response

(The authors gave the same response as above.)

Reviewer 3 Report
In this paper, the authors have designed a system, HomeMonitor, for monitoring artoiclend 20 controlling the behavior flow of smart home devices based on DESEND, an enhanced device event 21 signature extraction and detection method, which uses the packet size as the key feature and the 22 maximum packet interval as a mean to eliminate false positives. DESEND overcomes the problems 23 existing in the devices event signature-based methods and realizes the rapid detection of device 24 events in TCP and UDP data flows. The paper is well written and organized. However, it needs some improvements. The following are the suggestions to improve the article further:
- The introduction section has to be enhanced. A detailed discussion on the limitations of existing state of the art that motivated the current study has to be presented.
- Related works can be added to summarized in a table.
- Some of the the recent works such as the following can be discussed:
Eppda: An efficient privacy-preserving data aggregation federated learning scheme
- The results section can include some experimentations on the threat analysis on several kinds of attacks that can intrude the privacy of the data generated from iot devices.
- The authors have to present a comparative analysis on the results obtained in this study with state of the art.
- What is the computational complexity of the proposed approach?
- Discuss about the threats to validity of the proposed study.
- Present future enhancements of this work that can give directions to the researchers who want to extend and work on the future enhancements of this study.
Author Response

(The authors gave the same response as above.)

Round 2
Reviewer 1 Report
The manuscript can be accepted in this version.
Reviewer 2 Report
I think that the authors have well taken in consideration all remarks. Thus, I believe that this work can be accepted for publication in sensors journals
Reviewer 3 Report
All the comments are addressed